# Sodium, Potassium-Adenosine Triphosphatase as a Potential Target of the Anti-Tuberculosis Agents, Clofazimine and Bedaquiline

**DOI:** 10.3390/ijms252313022

**Published:** 2024-12-04

**Authors:** Khomotso Mmakola, Marissa Balmith, Helen Steel, Mohamed Said, Moliehi Potjo, Mieke van der Mescht, Nomsa Hlatshwayo, Pieter Meyer, Gregory Tintinger, Ronald Anderson, Moloko Cholo

**Affiliations:** 1Department of Immunology, Faculty of Health Sciences, University of Pretoria, Pretoria 0001, South Africa; khomotso.mmakola@tuks.co.za (K.M.); helen.steel@up.ac.za (H.S.); moliehi.potjo@nhls.ac.za (M.P.); mieke.vandermescht@tuks.co.za (M.v.d.M.); nomsa.hlatswayo@nhls.ac.za (N.H.); pieter.meyer@up.ac.za (P.M.);; 2Department of Pharmacology, Faculty of Health Sciences, University of Pretoria, Pretoria 0084, South Africa; marissa.balmith@up.ac.za; 3Department of Medical Microbiology, Faculty of Health Sciences, University of Pretoria, Pretoria 0001, South Africa; mohamed.said@up.ac.za; 4Department of Medical Microbiology, Tshwane Academic Division, National Health Laboratory Services, Pretoria 0001, South Africa; 5Department of Immunology, Tshwane Academic Division, National Health Laboratory Services, Pretoria 0002, South Africa; 6Department of Internal Medicine, Steve Biko Academic Hospital, Faculty of Health Sciences, University of Pretoria, Pretoria 0002, South Africa; greg.tintinger@up.ac.za; 7Clinical and Translational Research Unit of the Rosebank, Oncology Centre, Johannesburg 2196, South Africa; 8Basic and Translational Research Unit, Nuclear Medicine Research Infrastructure, Steve Biko Academic Hospital, Pretoria 0001, South Africa

**Keywords:** bedaquiline, clofazimine, cardiomyocytes, cellular viability, multidrug-resistant tuberculosis, sodium, potassium-adenosine triphosphatase, adenosine triphosphate

## Abstract

Multidrug-resistant tuberculosis (MDR-TB) patients are treated with a standardised, short World Health Organization (WHO) regimen which includes clofazimine (CFZ) and bedaquiline (BDQ) antibiotics. These two antibiotics lead to the development of QT prolongation in patients, inhibiting potassium (K^+^) uptake by targeting the voltage-gated K^+^ (Kv)11.1 (hERG) channel of the cardiomyocytes (CMs). However, the involvement of these antibiotics to regulate other K^+^ transporters of the CMs, as potential mechanisms of QT prolongation, has not been explored. This study determined the effects of CFZ and BDQ on sodium, potassium–adenosine triphosphatase (Na^+^,K^+^-ATPase) activity of CMs using rat cardiomyocytes (RCMs). These cells were treated with varying concentrations of CFZ and BDQ individually and in combination (1.25–5 mg/L). Thereafter, Na^+^,K^+^-ATPase activity was determined, followed by intracellular adenosine triphosphate (ATP) quantification and cellular viability determination. Furthermore, molecular docking of antibiotics with Na^+^,K^+^-ATPase was determined. Both antibiotics demonstrated dose–response inhibition of Na^+^,K^+^-ATPase activity of the RCMs. The greatest inhibition was demonstrated by combinations of CFZ and BDQ, followed by BDQ alone and, lastly, CFZ. Neither antibiotic, either individually or in combination, demonstrated cytotoxicity. Molecular docking revealed an interaction of both antibiotics with Na^+^,K^+^-ATPase, with BDQ showing higher protein-binding affinity than CFZ. The inhibitory effects of CFZ and BDQ, individually and in combination, on the activity of Na^+^,K^+^-ATPase pump of the RCMs highlight the existence of additional mechanisms of QT prolongation by these antibiotics.

## 1. Introduction

Tuberculosis (TB) disease is caused by the acid-fast bacterium, *Mycobacterium tuberculosis* (*M. tuberculosis*) and is currently the leading cause of morbidity and mortality worldwide due to a single infectious agent. One of the main contributing factors to the burden of the disease is an alarming increase in the number of multidrug-resistant TB (MDR-TB) cases. Recently, an efficient chemotherapeutic regimen containing two second-line drugs, namely clofazimine (CFZ) and bedaquiline (BDQ), has been introduced, shortening treatment schedules from 18–24 months to 9–12 months, as well as resulting in high treatment success rates (87–90% cure rates) [1,2,3,4].

However, despite these benefits in chemotherapy, CFZ and BDQ have been identified as the two main agents implicated in the development of prolongation between the start of the Q wave and end of the T wave on the electrocardiogram (QT) in patients, which is a risk factor for the development of cardiac arrhythmia, leading to cardiac arrest [2,3,5,6,7]. Previous studies have shown that both CFZ and BDQ lead to the development of QT prolongation by interfering with potassium (K^+^) influx of cardiomyocytes (CMs: cardiac muscle cells) [8,9,10,11,12] targeting the voltage-gated K^+^ (Kv) channel Kv11.1 (KCNH2: also known as the human Ether-a-go-go-related gene [hERG]) [7,13] and I_Kr_ [14]. In addition to the Kv11.1, CMs possess several other K^+^-uptake transporters, including the Kv1.3 (KCNA3), adenosine triphosphate (ATP)-sensitive potassium (IK_ATP_, also known as KCNJ11) channels [15,16,17], KCND2 (I_to1_), KCNQ1 (I_Ks_) [14], and the sodium, potassium–adenosine triphosphatase (Na^+^,K^+^-ATPase) pump [9,10,18,19], which participate in repolarization. However, the effects of both CFZ and BDQ on the activities of these transporters of the cardiac muscle cells have not been reported.

Despite the limited information on the other K^+^ transporters of CMs, CFZ, and BDQ have been shown to inhibit the Kv1.3 of thymus (T) lymphocytes [20] and monocytes [2,21,22,23]. In addition, CFZ has been shown to inhibit the Na^+^,K^+^-ATPase activity of T lymphocytes [24]. However, the effects of BDQ on the activity of the Na^+^,K^+^-ATPase in any type of mammalian cells have not been described.

In the current study, the effects of both antibiotics on the Na^+^,K^+^-ATPase activity of isolated CMs, as a potential in vitro mimic of the mechanism of QT prolongation, have been investigated. The Na^+^,K^+^-ATPase transporter is a transmembrane protein complex found in the cell membranes of all eukaryotic cells [25]. It is a member of the P-type ATPase class of enzymes, requiring adenosine triphosphate (ATP) as an energy source for its activity [26]. Its function is to maintain homeostasis of Na^+^ and K^+^ ion concentrations, as well as their electrochemical gradients across the plasma membrane [27], operating as an antiporter, transporting K^+^ inwardly in exchange for Na^+^ [28,29,30,31]. It contributes significantly to osmoregulation and maintenance of the resting membrane potential [31]. Cells use K^+^ for maintenance of the transmembrane potential (~90 millivoltage: mV) required for the activation of various metabolic activities, including protein synthesis, osmotic pressure, and regulation of pH (acid-base balance) [32,33,34].

In the current study, rat CMs (RCMs) were incubated in the absence and presence of the two antibiotics, both individually and in combination. Thereafter, cellular Na^+^,K^+^-ATPase activities were determined, followed by measurement of intracellular ATP concentrations and cellular viability. The molecular interactions of the antibiotics with Na^+^,K^+^-ATPase cellular protein interactions were analysed using computational molecular docking. Furthermore, to evaluate the effect of these antibiotics on additional K^+^ transporters, the interaction of these antibiotics with an additional transporter, an ATP-dependent channel, K_ATP_, was also examined using molecular docking.

## 2. Results

### 2.1. Confirmation and Antibiotic Treatments of the RCMs

The sizes and viabilities of the cells were confirmed using the scatter plot generated by the flow cytometer (Appendix A). Thereafter, the RCMs were treated with three concentrations of CFZ and BDQ antibiotics individually, prepared in double dilutions, which were 1.25, 2.5, and 5 mg/L, while the combination assays were treated with mixtures of the two antibiotics prepared at their mid-points (1.25C + 1.25B, 2.5C + 2.5B, and 5C + 5B; C, clofazimine; B, bedaquiline). Based on dimethylsulphoxide (DMSO) being used as a solvent for the antibiotic solutions, two controls were prepared, which included the DMSO-free and DMSO-treated control assays. The assays used included determination of the effects of the antibiotics on the Na^+^,K^+^-ATPase activity, intracellular ATP concentrations, and cellular viability. All antibiotic treatment assays were analysed using comparison with the corresponding DMSO-treated control systems.

### 2.2. Na^+^,K^+^-ATPase Activity of the RCMs

The Na^+^,K^+^-ATPase activity of individual antibiotic-free controls and antibiotic-treated samples was determined following the formula described in the Na^+^,K^+^-ATPase activity assay kit (Section 4.2.3). All the variables used in the determination of enzyme activity are as shown in Appendix A, Appendix A. The protein concentrations in the antibiotic-free controls and antibiotic-treated samples (C_pr_) are as shown in Appendix A while the Y intercept and the slope of the standard curve are in Appendix A, Appendix A. The results of the effects of the individual antibiotics on RCM Na^+^,K^+^-ATPase activity assays are as shown in Figure 1.

The mean Na^+^,K^+^-ATPase activities of the antibiotic-free controls were 8.04 ± 2.89 and 13.49 ± 1.8 µmol Pi/mg prot/hour for the DMSO-untreated (NDM: no DMSO) and -treated controls, respectively, showing an increase in Na^+^,K^+^-ATPase activity after treatment of cells with DMSO. The difference in activity between these controls was significant (*p* = 0.0087).

For antibiotic treatments, both test agents demonstrated inhibitory effects on the activity of Na^+^,K^+^-ATPase of the RCMs in a dose-dependent manner. Clofazimine demonstrated inhibitory effects on the Na^+^,K^+^-ATPase activity of the RCMs only at the highest concentrations of 2.5 and 5 mg/L, achieving statistically significant differences at both concentrations (*p* = 0.0043 and 0.0087, respectively). The inhibitory effects of these two CFZ concentrations resulted in 1.5-fold and 1.75-fold decreases in the Na^+^,K^+^-ATPase activity in relation to the controls, respectively. Interestingly, at 1.25 mg/L, CFZ demonstrated an increase in Na^+^,K^+^-ATPase activity, which was, however, not statistically significant (*p* = 0.24).

In the case of BDQ, unlike CFZ, all three concentrations of this agent demonstrated inhibition of RCM Na^+^,K^+^-ATPase activity. However, similarly to CFZ, BDQ achieved statistically significant inhibitory effects on enzyme activity only at the highest tested concentrations of 2.5 and 5 mg/L (*p* = 0.0152 and 0.0022, respectively). Exposure of RCMs to these concentrations of BDQ resulted in 1.4-fold and 2.2-fold levels of inhibition of Na^+^,K^+^-ATPase activity relative to the control values, respectively.

Similar to the effects of BDQ, treatment of the RCMs with all of the antibiotic combinations demonstrated inhibitory effects on Na^+^,K^+^-ATPase activity, achieving statistical significance at all combinations tested (*p* = 0.0022 for all concentrations). The inhibitory effects of these combination sets resulted in 2.1-fold, 2.2-fold, and 3.5-fold decreases in activity for 1.25C + 1.25B, 2.5C + 2.5B, and 5C + 5B (mg/L) concentrations in relation to the controls, respectively.

In summary, CFZ and BDQ, individually and in combination, demonstrated dose–response inhibition of the activity of Na^+^,K^+^-ATPase of the RCMs. Based on the magnitude of inhibition by the individual antibiotic assays, the highest inhibitory effects were demonstrated by combinations of the two antibiotics, followed by BDQ concentrations, while CFZ concentrations demonstrated lesser, albeit statistically significant, inhibitory effects on enzyme activity.

### 2.3. The Intracellular ATP Concentrations of the RCMs

Intracellular ATP concentrations in the different samples were determined (µmol/mL) as described in Section 4.2.3 using the formula, C = B/V × D, where C is sample ATP concentration [nanomoles (µmol/mL)]; B, the ATP concentration in each sample extrapolated using OD measurements from the standard curve (Appendix A), Appendix A; V, the volume of sample in reaction well (0.015 mL) while D is a dilution factor, which remains 1 as the samples were used undiluted. The final ATP concentrations are as shown in Figure 2.

For the antibiotic-free control systems, the intracellular concentrations of ATP were 5.67 ± 0.34 and 5.48 ± 0.24 µmol/mL for the DMSO-untreated and -treated samples, respectively, showing a decline in ATP concentrations following treatment of cells with DMSO, achieving a statistically significant difference (*p* = 0.0102).

For the antibiotic treatments, all antibiotic-treated assays demonstrated increases in ATP concentrations. Clofazimine demonstrated a dose–response increase in ATP concentrations, reaching peak levels at 5 mg/L CFZ. However, the increase in ATP levels was significantly different at all antibiotic concentrations tested (1.25–5 mg/L) (*p* < 0.05).

Similarly to CFZ, BDQ demonstrated dose–response increases in intracellular ATP concentrations, which were significantly different from the controls at all concentrations tested, being maximal at 1.25 mg/L in the case of BDQ.

Similarly to the individual antibiotics, combinations of the test antibiotics also demonstrated increased intracellular ATP concentrations. However, in contrast to the individual antibiotics, dose–response increases in ATP concentrations were detected only with the first two antibiotic combination sets, viz. 1.25C + 1.25B and 2.5C + 2.5B mg/L and similar to BDQ, being maximal at 1.25C + 1.25B mg/L.

These results illustrate that treatment of cells with CFZ and BDQ individually or in combination resulted in increases in intracellular ATP concentrations, presumably due to inhibition of Na^+^,K^+^-ATPase.

### 2.4. Viability of the RCMs

Cellular viabilities of antibiotic-free controls and -treated cells were determined by flow cytometry using propidium iodide (PI) straining method. The numbers of viable cells in all the assays were determined as percentages in relation to the DMSO-treated controls (Figure 3).

The results showed that treatment of cells with DMSO did not significantly change the number of viable cells. Furthermore, treatment of cells with the two antibiotics, either individually or in combination, did not change the number of viable cells, resulting in cellular viabilities ranging between 95 and 100%.

### 2.5. Molecular Docking Analysis

#### 2.5.1. Antibiotic-Na^+^,K^+^-ATPase Interaction

The binding affinities of the two antibiotics, CFZ and BDQ, on Na^+^,K^+^-ATPase were evaluated individually using molecular docking to assess their potential as inhibitors of the Na^+^,K^+^-ATPase protein. The results showing the docking scores and the protein–ligand interactions are as shown in Figure 4.

The docking scores, expressed in kcal/mol, represent the strength of interaction between each antibiotic and the active site of the Na^+^,K^+^-ATPase protein, where more negative values indicate stronger binding. Clofazimine displayed a docking score of −3.686 kcal/mol, indicating moderate binding affinity to the Na^+^,K^+^-ATPase protein. This suggests that CFZ can interact with the binding pocket of the target, although with relatively lower affinity when compared to BDQ. In contrast, BDQ exhibited a more favourable docking score of −4.499 kcal/mol, indicating stronger binding affinity to the Na^+^,K^+^-ATPase protein than CFZ. This enhanced binding affinity suggests that BDQ may form more stable interactions with the active site, potentially making it a more effective inhibitor of the Na^+^,K^+^-ATPase protein in comparison to CFZ. These docking results suggest that BDQ has a stronger potential for inhibitory action on the Na^+^,K^+^-ATPase protein than CFZ.

The ligand interaction diagram depicts the binding of CFZ within the active site of the Na^+^,K^+^-ATPase protein, showing several important interactions that stabilise the binding (Figure 4a). Clofazimine is stabilised through hydrogen bonds, notably with aspartate (ASP)450, which plays a crucial role in anchoring the ligand via electrostatic interactions. Additionally, a halogen bond is formed between the chlorine atom on CFZ and arginine (ARG)692. A pi-cation interaction can be seen with ARG551. Hydrophobic contacts are evident, with residues like leucine (LEU)553, phenylalanine (PHE)482, and alanine (ALA)510. These interactions likely contribute to the moderate binding affinity reflected in the docking score of −3.686 kcal/mol for CFZ.

The ligand interaction diagram illustrates the binding mode of BDQ in the active site of the Na^+^,K^+^-ATPase protein (Figure 4b). Interactions are evident between the ligand, serine (SER)694, and ASP376. Hydrophobic interactions play a significant role in the binding affinity, involving residues like PHE482, valine (VAL)616, and VAL719. The combination of these interactions results in the relatively strong binding affinity observed, supported by a docking score of −4.499 kcal/mol for BDQ.

#### 2.5.2. Antibiotic-K_ATP_ Channel Interaction

Similar to the Na^+^,K^+^-ATPase protein, molecular docking was performed to evaluate the binding affinities of the two antibiotics, CFZ and BDQ, individually to the sulfonylurea receptor (SUR)2A (PDB ID: 7Y1J) subunit of the K_ATP_ protein in order to assess their potential inhibitory activity on this protein. The stability and the binding strength of the antibiotics within the active site of the K_ATP_ protein were assessed. The results showing the docking score and the protein–ligand interaction are as shown in Figure 5.

CFZ demonstrated a docking score of −5.294 kcal/mol, whereas BDQ achieved a docking score of −7.729 kcal/mol, suggesting a stronger binding affinity to the SUR2A subunit than CFZ.

The ligand interaction diagram depicts the binding of CFZ within the active site of the SUR2A protein (Figure 5a). In this diagram, ARG1213 forms a pi-cation interaction with the aromatic ring on the ligand. Additionally, pi-pi stacking is seen between tryptophan (TRP)1260 and the ligand. Hydrophobic contacts are also significant in this binding. These involve residues such as tyrosine (TYR)370, PHE426, LEU427, LEU584, and TYR1205. These hydrophobic interactions likely played a major role in securing the ligand within the hydrophobic pocket of the protein. These interactions collectively supported the binding stability of CFZ within the active site of the SUR2A protein, reflecting its moderate binding affinity.

The ligand interaction diagram depicting the binding mode of BDQ in the active site of the protein is as shown in Figure 5b. Pi-cation interactions were present with ARG1213, where the positively charged arginine residue interacts with the ligand’s aromatic ring. A salt bridge interaction with glutamic acid (GLU)1216 provides additional stability to the ligand’s position within the active site. Additionally, pi-pi stacking can be seen between TRP1260 and the aromatic ring on the ligand. Furthermore, a halogen bond formed between ARG1112 and bromine (Br) on the ligand. The diagram also highlights hydrophobic interactions with several residues, including LEU366, LEU427, TYR370, PHE426, and TRP423. These interactions likely allowed the ligand to fit snugly within the hydrophobic pocket of the protein, further stabilising its binding. These results demonstrate that BDQ has a stronger binding affinity to the K_ATP_ subunit than CFZ.

## 3. Discussion

The application of the multidrug, short-course, MDR-TB-treatment regimen has led to several chemotherapeutic successes in the management of MDR-TB patients [2]. This treatment regimen is administered over a shorter period of 9–12 months, as opposed to 18–24 months with the previous longer MDR-TB treatment regimen. It has also led to an improvement in treatment outcomes, increasing treatment success rates from 30–55% to 87–90% [1,6]. More importantly, it is associated with low relapse rates in patients [1,2,4,35].

Despite its beneficial effects, this treatment regimen is also associated with development of adverse events in patients, which differ in severity, ranging from mild to fatal [2]. One of the most severe adverse events is QT prolongation, which leads to fatal cardiac arrhythmia [36,37,38,39]. Cardiac arrhythmia develops as a result of damage to the cardiac muscle, affecting contraction, resulting in abnormal heart rates [36,37]. Among the constituent antibiotics of the MDR-TB short-course treatment regimen, CFZ and BDQ have been identified as the main antibiotics associated with development of QT prolongation, despite being associated with beneficial therapeutic antimicrobial effects [2,13,40,41,42,43,44,45].

Previous studies have demonstrated that the mechanism of QT prolongation by CFZ and BDQ is via inhibition of K^+^ uptake through blockage of the voltage-gated K^+^-uptake channel, Kv11.1 of CMs [13,46]. However, information on the involvement of these antibiotics on other K^+^-uptake transporters of CMs, including the K^+^ channels and the Na^+^,K^+^-ATPase transporters, in the mechanisms of QT prolongation has not been described.

In the current study, the potential inhibitory effects of these antibiotics on the Na^+^,K^+^-ATPase of the CMs were investigated. Due to challenges in accessing human CMs (HCMs), we used the RCMs in place of the HCMs. The effects of the antibiotics individually and in combination on the Na^+^,K^+^-ATPase activity of the RCMs were investigated by determining the rates of ATP decomposition by measuring the rate of intracellular Pi release in the absence and presence of the test antibiotics. Based on the utilisation of ATP for the activity of the Na^+^,K^+^-ATPase pump [30,47], the amounts of intracellular ATP were also determined by using a spectrophotometric procedure. Furthermore, the cytotoxic effects of these antibiotics on the RCMs were evaluated by viability determination using PI-based staining flow cytometry. We further evaluated the interactions of these antibiotics with the Na^+^,K^+^-ATPase protein individually using computational molecular docking. In order to determine the effects of these antibiotics on additional K^+^ transporters, the interaction of the antibiotics with the K_ATP_ channel, targeting the SUR2A subunit, was assessed using molecular docking.

The results of the current study showed that both CFZ and BDQ, individually and in combination, have inhibitory effects on the activity of the Na^+^,K^+^-ATPase of RCMs. However, the degree of inhibition by these antibiotics differed. Clofazimine demonstrated inhibitory effects on the Na^+^,K^+^-ATPase activity only at higher concentrations of 2.5 and 5 mg/L, which were also statistically significant. These inhibitory effects of higher concentrations of CFZ on Na^+^,K^+^-ATPase activity coincided with statistically significant elevations in cellular ATP concentrations, illustrating decreased utilisation of ATP as a consequence of attenuation of the activity of the Na^+^,K^+^-ATPase transporter, highlighting the Na^+^,K^+^-ATPase transporter as being a significant consumer of cellular ATP. Despite exhibiting this inhibitory effect on Na^+^,K^+^-ATPase activity, CFZ did not demonstrate any significant change in cellular viability, demonstrating an absence of cytotoxicity, showing that the inhibition in enzymatic activity is not dependent of the number of viable cells.

Unlike CFZ, BDQ demonstrated a dose-related inhibitory effect on the Na^+^,K^+^-ATPase of the RCMs at all concentrations tested, indicating that BDQ is more potent than CFZ with respect to inhibition of this enzyme. However, like CFZ, BDQ achieved significant inhibitory effects on the enzyme activity at the highest concentrations of 2.5 and 5 mg/L, which coincided with elevated levels of ATP at all BDQ concentrations tested. However, the ATP concentrations declined somewhat at these highest concentrations of BDQ (2.5 and 5 mg/L), suggesting the simultaneous presence of inhibitory effects of BDQ on ATP synthesis. Despite the defective ATP synthesis coupled with inhibition of Na^+^,K^+^-ATPase activity, similar to CFZ, BDQ did not show any significant effect on the cellular viability, demonstrating an absence of cytotoxicity on the RCMs in the current experimental settings.

The most potent inhibitory effects of the antibiotics on the Na^+^,K^+^-ATPase activity of the RCMs were observed with the antibiotic combination assays, which, unlike the individual antibiotics, attained statistical significance at all combinations of concentrations tested (1.25C + 1.25B to 5C + 5B mg/L). Inhibition of enzymatic activity was associated with increased intracellular ATP levels, which were dose-related, with the highest ATP concentrations being detected at the lowest concentrations of the tested antibiotic combinations (1.25C + 1.25B mg/L), declining at higher concentrations of combinations in a dose-dependent manner. The inhibitory effects of the combination assays on the ATP concentrations may be attributable to the presence of BDQ, as it demonstrated similar effects of attenuating ATP levels at higher concentrations on the RCMs when operating alone (Figure 2). Similar to the individual antibiotics, despite a high degree of attenuation of Na^+^,K^+^-ATPase activity, the inhibitory effects of the antibiotic combinations did not lead to cellular cytotoxicity, but possibly led to insignificant damage to the CMs, even when the antibiotics were used in combination during antimicrobial chemotherapy. However, this may differ with prolonged treatment with CFZ and/or BDQ as opposed to the short incubation times used in the current in vitro study.

Although the current experimental data were generated on RCMs, which differ from human CMs, the inhibitory effects of the two antibiotics on Na^+^,K^+^-ATPase activity were confirmed through binding of these antibiotics at the E1.3Na.ATP conformational state of the human Na^+^,K^+^-ATPase protein individually using molecular docking analysis [48]. Similarly, the stronger binding affinity of BDQ to the Na^+^,K^+^-ATPase protein than CFZ further confirms its higher inhibitory effect on the transporter than CFZ. This, however, may not be the only mechanism by which CFZ and BDQ interfere with the activity of Na^+^,K^+^-ATPase. In this context, it is noteworthy that both antibiotics are cationic amphiphiles which possess membrane-destabilizing properties, which are known to attenuate the functioning of membrane ion transporters.

The impact of the inhibitory activity of CFZ and BDQ on other K^+^ transporters was further evaluated by determining the interaction of the antibiotics with the K_ATP_ K^+^ efflux channel [16]. The K_ATP_ channel is inhibited by the presence of high intracellular ATP, while it is activated by high concentrations of adenosine diphosphate (ADP) [49]. Molecular docking results performed in the current study, on the interaction of the two antibiotics with the SUR2A molecule of the cardiac K_ATP_ protein [49,50], showed that both antibiotics had high affinities for the K_ATP_ protein, with BDQ showing stronger affinity for the protein than CFZ, implicating the K_ATP_ as a possible additional cellular target of the two antibiotics in their involvement in QT prolongation. The SUR2A-containing K_ATP_ channel in the heart remains closed in normal conditions and only opens in response to severe metabolic inhibition, such as ischemic stress [49]. This compensatory opening of K_ATP_ channels to the Na^+^,K^+^-ATPase may contribute to cardioprotection [16,49,51].

Despite these findings, the functional effects of our findings, specifically effects on mitochondria and glycolytic activity, with their roles in ATP synthesis, as well as on the influence of other essential K^+^ channels, remain to be elucidated. Although not yet demonstrated in human cardiac muscle cells, both antibiotics have been shown to increase glycolysis in other types of human cells, such as human monocyte-derived macrophages (hMDM), while they have failed to show any effect on the oxidative phosphorylation system (OXPHOS) of the same cells [52]. However, CFZ demonstrated reduced ATP production through an increase in proton leak and reduced mitochondrial uncoupling in these cells [52]. It has also been described that intracellular K^+^ depletion does indeed attenuate mitochondrial function [53].

The current study is the first to report the inhibitory effects of both CFZ and BDQ individually and in combination on the Na^+^,K^+^-ATPase of CMs, which is a potential augmentative mechanism of the development of QT prolongation. Previous studies have demonstrated that the development of prolonged QT intervals [45,46,54] as a result of attenuation of Na^+^,K^+^-ATPase of the CMs caused by other drugs is associated with cardiotoxicity [31,55]. However, in the current study, the inhibition of the Na^+^,K^+^-ATPase activity in the absence of cytotoxicity may not exclude the involvement of these two antibiotics in cardiotoxicity, but may be due to the experimental design, which involved a short 15 min exposure time of the antibiotics to the cells, as mentioned above. Additionally, the discrepancy might be a result of the use of RCMs, which differ in characteristics from HCMs [56,57].

Nonetheless, the findings of the current study highlight attenuation of Na^+^,K^+^-ATPase activity as a potential novel mechanism by which CFZ and BDQ contribute to the development of QT prolongation during anti-tuberculosis chemotherapy, leading to cardiac arrhythmia.

In conclusion, in addition to previous findings which identified the Kv11.1 K^+^-uptake channels of the CMs as targets of CFZ and BDQ, the results of the current study have also identified the Na^+^,K^+^-ATPase of the CMs as a potential target of these antibiotics, suggesting inhibitory effects of the antibiotics on the Na^+^,K^+^-ATPase as possible contributors to the development of QT prolongation in TB patients. These findings are indicative of possibly shared, as well as distinct, molecular/biochemical mechanisms of CFZ- and BDQ-mediated interference with the activity of Na^+^,K^+^-ATPase in the development of cardiac dysfunction, which remain to be elucidated. These results therefore highlight the necessity of a careful selection of antibiotics in future design of short-course MDR-TB regimens.

## 4. Materials and Methods

### 4.1. Materials

#### 4.1.1. Antibiotics and Other Chemicals

The antibiotics investigated in the current study were CFZ and BDQ. Clofazimine was purchased from Sigma Aldrich (Sigma Chemical Co., St. Louis, MO, USA), while BDQ was obtained from Adooq Bioscience (York, UK). The antibiotics were dissolved in DMSO and used at final concentrations of 1.25–5 mg/L (0.002–0.01 mM for both antibiotics), individually and in combination, in all of the experimental assays. The DMSO was used at a final concentration of 0.1% for solvent control in all the control systems.

Unless otherwise stated, all other chemicals and reagents were purchased from Sigma-Aldrich, Lasec (Cape Town, RSA) and Whitehead Scientific (Cape Town, RSA). Propidium iodide (PI) dye [deoxyribonucleic acid (DNA) Prep Stain], used for determining cell viability, was purchased from Beckman Coulter (Brea, CA, USA).

#### 4.1.2. Assay Kits

The assay kits used in the current study included the Na^+^,K^+^-ATPase activity kit (Elabscience Biotechnology Inc., Houston, TX, USA) for determination of Na^+^,K^+^-ATPase activity, as well as the ATP colorimetric assay kit (Abcam, ab282930) (Abcam, Waltham, MA, USA) for quantification of cellular ATP concentrations.

#### 4.1.3. RCMs and Growth Media

Rat cardiomyocytes (RCMs, neonatal rat ventricular cardiomyocytes, [Catalog No. R-CM-561]) were sourced from Lonza Bioscience (Walkersville Inc., Walkersville, MD, USA). They were cryopreserved and stored at −196 °C in liquid nitrogen.

The RCM growth medium (RCGM) was used for preparation of the RCMs. This was prepared by using the RCGM bullet Kit (200 mL of RCM basal medium (RCBM) and the RCGM SingleQuots kit) (Lonza Bioscience, Walkersville, MD, USA). The RCGM was prepared by mixing the ingredients of the RCGM SingleQuots kit (300 µL of antibiotic mixture: gentamycin sulphate and amphotericin B, 18 mL of horse serum from Equidae, and 18 mL of foetal bovine serum) with 200 mL of the RCBM.

### 4.2. Methods

#### 4.2.1. Preparation of the RCMs

A vial containing 1 mL of RCM cells was taken from liquid nitrogen storage, and the cells were thawed by gentle rubbing of the vial with fingers for approximately two–three minutes. The cells were then transferred into a 15 mL tube, RCGM medium was added stepwise, and the cell suspension was mixed by gently inverting the 15 mL tube. The cell suspension was then incubated at 37 °C for 3.5 h in the presence of 5% CO_2_. The RCGM medium was removed from the cells by centrifugation at 335× *g* and 4 °C for 10 min, and the supernatant was discarded thereafter. The cell pellet was resuspended in pre-warmed saline solution (0.9%, NaCl: sodium chloride). The characteristics of the cells with respect to size and viability were determined by flow cytometry by mixing 50 µL of cell suspension with 450 µL of propidium iodide, and the mixture was analysed using a Beckman Coulter Navios EX Flow Cytometer (Beckman Coulter, Maryfort, O’Callaghans Mills, Co. Clare, Ireland), using Navios EX Software version 2.2.

#### 4.2.2. Antibiotic Reaction Assays

Approximately 3 × 10^5^ cells/mL were transferred into different 1.5 mL Eppendorf tubes. Thereafter, 1 µL samples of varying stock concentrations of antibiotics (CFZ, BDQ and their combinations) were added to the cells. The antibiotic concentrations were prepared in double dilutions ranging from 1.25 to 5 mg/L for both CFZ and BDQ. For combination assays, the individual antibiotics were mixed at their midpoint concentrations [e.g., ½ (1.25 mg/L CFZ) + ½ (1.25 mg/L BDQ)]. In reaction mixtures, the concentrations for each antibiotic or their combinations are shown by a number and symbol, with C, B, or C + B representing CFZ, BDQ individually, or their combination, respectively, such as 1.25C, 1.25B, or 1.25C + 1.25B. Two antibiotic-free controls, solvent-free (no DMSO) and DMSO-treated, prepared by adding 1 µL of 100% DMSO yielding 0.1% DMSO, were included. The suspensions were mixed by inverting the tubes, and the mixtures were incubated at 37 °C for 15 min in the presence of 5% CO_2_. The reactions were stopped by placing the tubes on ice for one minute, and the contents were analysed for Na^+^,K^+^-ATPase activity, ATP quantification, and cellular viability.

#### 4.2.3. Na^+^,K^+^-ATPase Activity

The enzymatic activity of Na^+^,K^+^-ATPase was determined according to the method of Chen et al. [58], using the Na^+^,K^+^-ATPase activity kit (Section 4.1.2), following the manufacturers’ instructions. Detection of enzymatic activity was based on Na^+^,K^+^-ATPase-mediated decomposition of ATP to produce ADP and Pi. The amount of Pi was then quantitated and used to calculate the Na^+^,K^+^-ATPase activity. The ADP and Pi were extracted from the cells as protein constituents.

##### Protein Extraction

Protein samples used for measurement of Na^+^,K^+^-ATPase enzymatic activity were extracted from the control and antibiotic-treated RCMs using the vortexing method. Following centrifugation at 335× *g* at 4 °C for 10 min, the supernatants were discarded and the pellets were vortexed, followed by the addition of 500 µL saline solution. Approximately 500 µL of beads (2 mm diameter) was added to each cell suspension, and the mixtures were vortexed for a maximum period of five minutes, with cooling on ice every two minutes. The cell suspensions were then centrifuged at 335× *g* at 4 °C for one minute, and approximately 500 µL of the supernatants, containing the protein extracts, was harvested. Protein concentrations of the samples were determined at 280 nm using a NanoDrop spectrophotometer (NanoDrop Technologies, Wilmington, DE, USA) and recorded as mg/mL.

##### Na^+^,K^+^-ATPase Enzymatic Activity Assay

The protein extracts were also used for the determination of Na^+^,K^+^-ATPase activity. The Na^+^,K^+^-ATPase reactions were performed using the method based on the colorimetric detection of the released free Pi.

The reaction mixture tubes were incubated at 37 °C for 10 min. Thereafter, protein precipitator reagent was added to precipitate the proteins to promote the release of free Pi. The tubes were vortexed and centrifuged at 8000× *g* for 10 min at 4 °C, and the supernatants were harvested for colorimetric detection of Pi.

Reaction mixtures for each sample (control and antibiotic treated) were prepared by mixing 20 µL of each sample with 200 µL of chromogenic working solution. Standard reaction mixtures (containing known standard Pi concentrations) were also prepared by mixing individual standards with chromogenic working solutions. The reaction mixtures were incubated at 37 °C for 15 min, followed by optical density (OD) measurements at 660 nm using a PowerWave_x_ spectrophotometer (Bio-Tek Instruments, Inc., Winooski, VT, USA). The optical densities (ODs) of the standard Pi concentrations were used for generation of the standard curve.

The Na^+^,K^+^-ATPase activities for each sample were calculated using the following formula:Na^+^,K^+^-ATPase activity (µmol Pi/mg protein/hour) = (ΔA_660_ − b)/a × V_1_/(C_pr_ × V_2_)/t × f,
where:y: OD_standard_ − OD_blank_ (OD_blank_ is the OD value when the standard concentration is 0);x: the concentration of the standard;ΔA_660_: OD_sample_ − OD_control_;b: the y intercept of the standard curvea: the slope of the standard curve;V_1_: the total volume of the reaction system (0.25 mL);C_pr_: the concentration of protein in sample (mg protein/mL);V_2_: the volume of added sample (0.1 mL);t: the time of the enzymatic reaction (1/6 h);f: the dilution factor of the sample before testing.

#### 4.2.4. ATP Concentration Determination

Quantification of ATP was determined using the ATP colorimetric assay kit (Abcam, ab282930: Section 4.1.2). The assay is based on phosphorylation of glycerol (transfer of phosphate group to glycerol from ATP) using glycerol kinase, producing ADP and glycerol phosphate. The procedure involves the extraction of ATP from lysed cells, followed in this case by deproteinization and, ultimately, quantification of ATP.

##### ATP Extraction

The control and antibiotic-treated reaction mixtures were centrifuged at 4833× *g* for 10 min at 4 °C. The supernatants were discarded and the pellets vortexed. Thereafter, 50 μL of ice-cold ATP assay buffer (supplied in the kit) was added and vortexed to lyse the cells, followed by centrifugation of the mixtures at 4833× *g* for 10 min at 4 °C. About 100 µL of each lysate was harvested and transferred into separate 1.5 mL Eppendorf tubes. These samples were placed on ice immediately to prevent consumption and utilisation of ATP by enzymes present in the samples. The proteins (and enzymes) were then removed from the supernatant lysates by a deproteinisation procedure performed using a spin column method. A 100 μL volume of the lysate of each sample was transferred immediately onto a spin column (Qiagen RNeasy mini kit: Qiagen GmbH, Hilden, DE, Germany), which was placed in a new 1.5 mL Eppendorf tube. The tubes were centrifuged immediately at 4833× *g* for 10 min at 4 °C, and the eluates were collected.

##### ATP Quantification

The deproteinated samples were used thereafter for ATP concentration determination. The concentrations of ATP in the eluate samples were determined using the ATP reaction assay kit following the manufacturers’ instructions, with no modification. Briefly, the deproteinated samples and the standard ATP concentrations samples were used for preparation of reaction mixtures in a 96-well plate to final volumes of 0.015 mL/ well. The 96-well plate was then incubated for 45 min at room temperature in the dark, and the ODs of the contents of the wells were determined at 570 nm using a PowerWave_x_ spectrophotometer (Bio-Tek Instruments, Inc., Winooski, VT, USA). The OD values of the standard concentrations were used to plot the standard curve, which was used to determine the ATP concentrations in the samples.

The ATP concentration for each sample was determined using the formula:Sample ATP concentration [C, nanomoles (µmol)/mL] = B/V × D
where:B = ATP concentration in the reaction well from standard curve (µmol);V = the sample volume added into sample wells (mL);D = the dilution factor.

#### 4.2.5. Cellular Viability Determination

To determine the cytotoxic potential of the test antibiotics, the viability of the cells in the antibiotic-treated and control reaction mixtures (Section 4.2.2) was determined by using the PI staining method. The PI dye bound to the nuclei of lysed cells, forming a nuclear–PI complex, which fluoresced, but did not bind to live cells, which it failed to penetrate.

A 50 μL aliquot of each sample, either the control or various antibiotic-treated systems (Section 4.2.2), was transferred into a 5 mL flow tube, and 450 µL of PI reagent (DNA Prep Stain: Beckman Coulter Life Sciences, Beckman Coulter, Inc., Indianapolis, IN, USA) was added. The mixture in the flow tube was vortexed and inserted immediately into the Navios EX Flow Cytometer instrument (Beckman Coulter, Maryfort, O’Callaghans Mills, Co. Clare, Ireland), and viability was determined according to the resultant scatter graph analysed using Navios EX Software version 2.2. The dead and live cells were located in different positions on the scatter chart, resulting in positioning of live cells closer to zero on the X-axis, while dead cells were positioned further away from zero in the positive direction.

#### 4.2.6. Computational Molecular Docking Analysis

The interaction between ligands and proteins was determined using computational molecular docking analysis.

##### Ligand Preparation

The chemical structures of CFZ and BDQ were downloaded from the NCBI PubChem database (https://pubchem.ncbi.nlm.nih.gov, accessed on 21 October 2024 and 11 November 2024) and saved in a 3D SDF format. Ligands were prepared using the Ligprep interface in Schrodinger Maestro (Schrödinger Release 2023-2: Maestro, version 13.3: New York, NY, USA). An OPLS_2005 force field was used at pH 7.4 ± 1 using Epik (Schrödinger Release 2023-2: Maestro, version 13.3). Desalt and generate tautomers were also selected on the LigPrep interface, and the stereoisomer computation was selected to retain specific chiralities and to generate, at most, three variations per ligand. Further refinement was performed using the LigPrep module in Maestro to preserve the chirality and original ionisation states. Tautomers were generated to account for different ligand forms. The Monte Carlo method was employed to explore the conformational space by rotating all single bonds, ensuring flexibility. This process continued until the global energy minimum was identified multiple times, confirming the most stable conformation. Energy minimisation was conducted using the OPLS_2005 force field, applying least-squares minimisation to ensure accurate structural refinement. This comprehensive approach ensured the ligands were optimised with the lowest energy, best conformers, and appropriate molecular properties for molecular docking studies.

##### Protein Preparation

Protein molecules

Two protein molecules, the Na^+^,K^+^-ATPase and the K_ATP_ channel, were assessed for ligand–protein interaction in the current study. The Na^+^,K^+^-ATPase protein structure, at E1.3Na.ATP conformational state, was retrieved from the Protein Data Bank (PDB ID: 7E21) (available from: https://www.rcsb.org, accessed on 21 October 2024) [48].

The cardiac muscle K_ATP_ channel consists of the inward rectifying K^+^ (Kir)6.2 and the SUR2A subunits [16,49,50]. The Kir6.2 contains the ATP-binding site inhibited by ATP, while the SUR2A is an ATP-binding cassette (ABC) protein, which, however, lacks membrane transport activity but functions in regulating the expression and activity of the K_ATP_ protein. The SUR2A protein is also regulated by Mg-nucleotides. Based on its activity requirement, the inward-facing (IF) conformation of SUR2A_IF/MgATP/MgATP_ structure stabilised by high concentration of ATP molecules retrieved from PDB ID: 7Y1J, also available at https://www.rcsb.org (accessed on 11 November 2024) [49], was used.

2.Protein preparation

Protein preparation was carried out using the Protein Preparation Wizard in Maestro. During this process, water molecules were removed, bond orders were assigned, hydrogens were added to the structure, and bonds to metal ions were deleted, with adjustments made to the formal charges of the metal and nearby atoms within a defined radius. The protonation states of the residues were then adjusted to reflect physiological conditions, correcting any potential structural errors. A restrained minimisation was applied to reduce steric clashes and optimise the geometry while preserving the overall protein conformation. A grid-based cavity prediction algorithm was employed to identify potential binding sites. The OPLS-2005 force field was used for energy minimisation throughout the preparation process.

##### Active Site Determination and Receptor Grid Generation

The receptor grid generation tool in Schrodinger Maestro version 13.3 was used to generate the grid (Schrödinger Release 2023-2: Maestro, version 13.3: New York, NY, USA). Since the proteins had an existing co-crystal structures [48,49], the receptor grid was determined by selecting the existing ligand.

##### Molecular Docking

The docking procedures were conducted using the Schrodinger Maestro version 13.3 Glide tool (Schrödinger Release 2023-2: Maestro, version 13.3: New York, NY, USA). The prepared antibiotic ligands were docked to the target proteins using the extra precision (XP) mode.

### 4.3. Statistical Analysis

Statistical analyses were performed on all data using GraphPad InStat Programme version 3.0 (GraphPad Software, San Diego, CA, USA). The results of each series of experiments were expressed as the mean values ± standard deviations (SDs). Due to all antibiotic mixtures prepared in DMSO, statistical significance was calculated between antibiotic-treated and DMSO-treated control systems using the Mann–Whitney U-test for comparison of non-parametric data. A *p* value of ≤ 0.05 was considered significant.

## Figures and Tables

**Figure 1 ijms-25-13022-f001:**
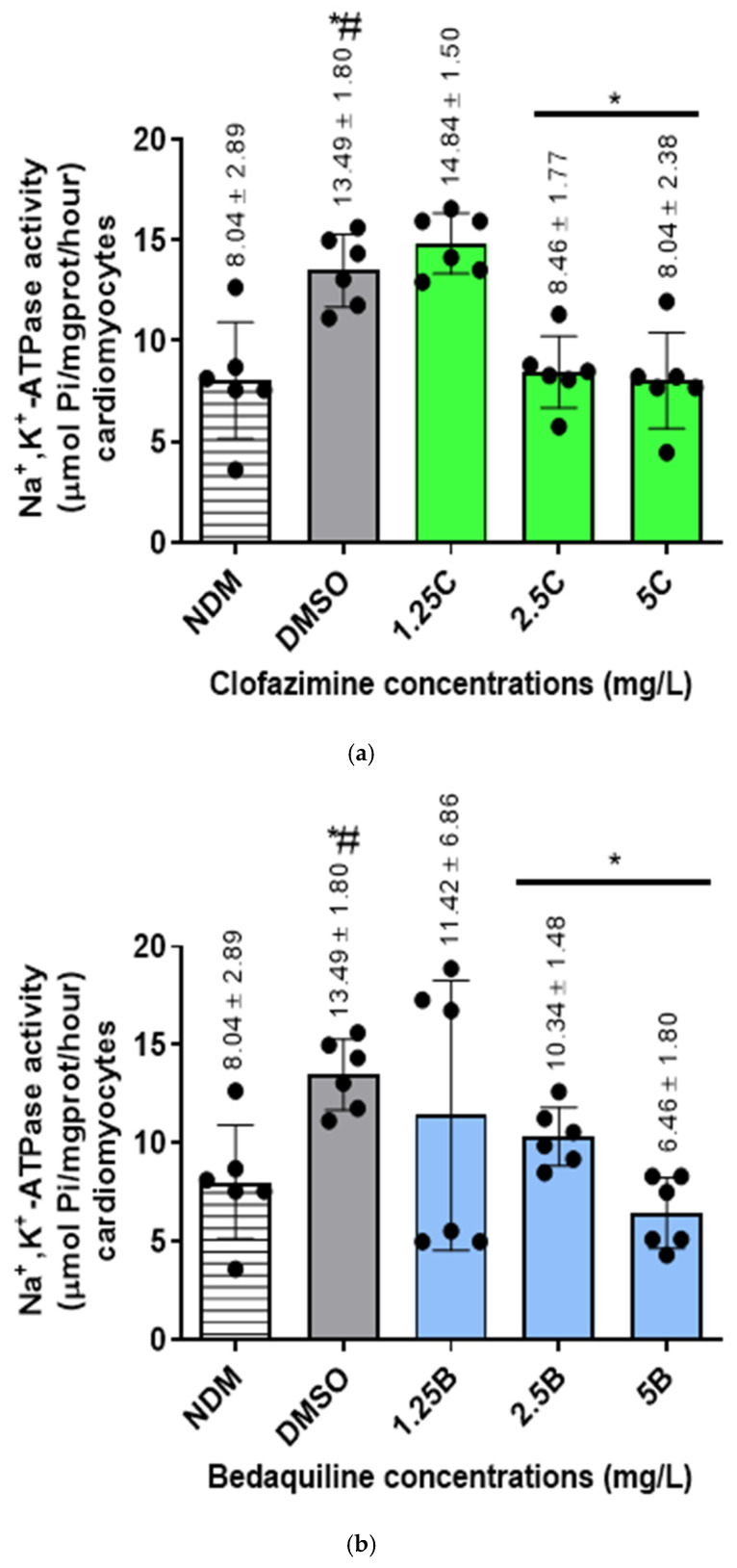
The Na^+^,K^+^-ATPase activities (µmol Pi/mg prot/hour) of the RCMs treated with varying concentrations of (**a**) CFZ and (**b**) BDQ alone and (**c**) in combination. Statistical significance, representing *p* ≤ 0.05, is shown by *. The *p* value representing statistical difference between NDM and DMSO antibiotic-free controls is denoted by #. The *p* value determined between the NDM and DMSO-treated controls was 0.0087. For the antibiotic treatment assays, the *p* values were: for CFZ = 0.24, 0.0043, and 0.0087 for 1.25C, 2.5C, and 5C; for BDQ = >0.99, 0.0152, and 0.0022 for 1.25B, 2.5B, and 5B; for CFZ and BDQ combinations = 0.0022 for all combinations. Abbreviations: B, bedaquiline; C, clofazimine; DMSO, dimethylsulphoxide; NDM, no DMSO; Na^+^,K^+^-ATPase, sodium, potassium–adenosine triphosphatase; RCMs, rat cardiomyocytes.

**Figure 2 ijms-25-13022-f002:**
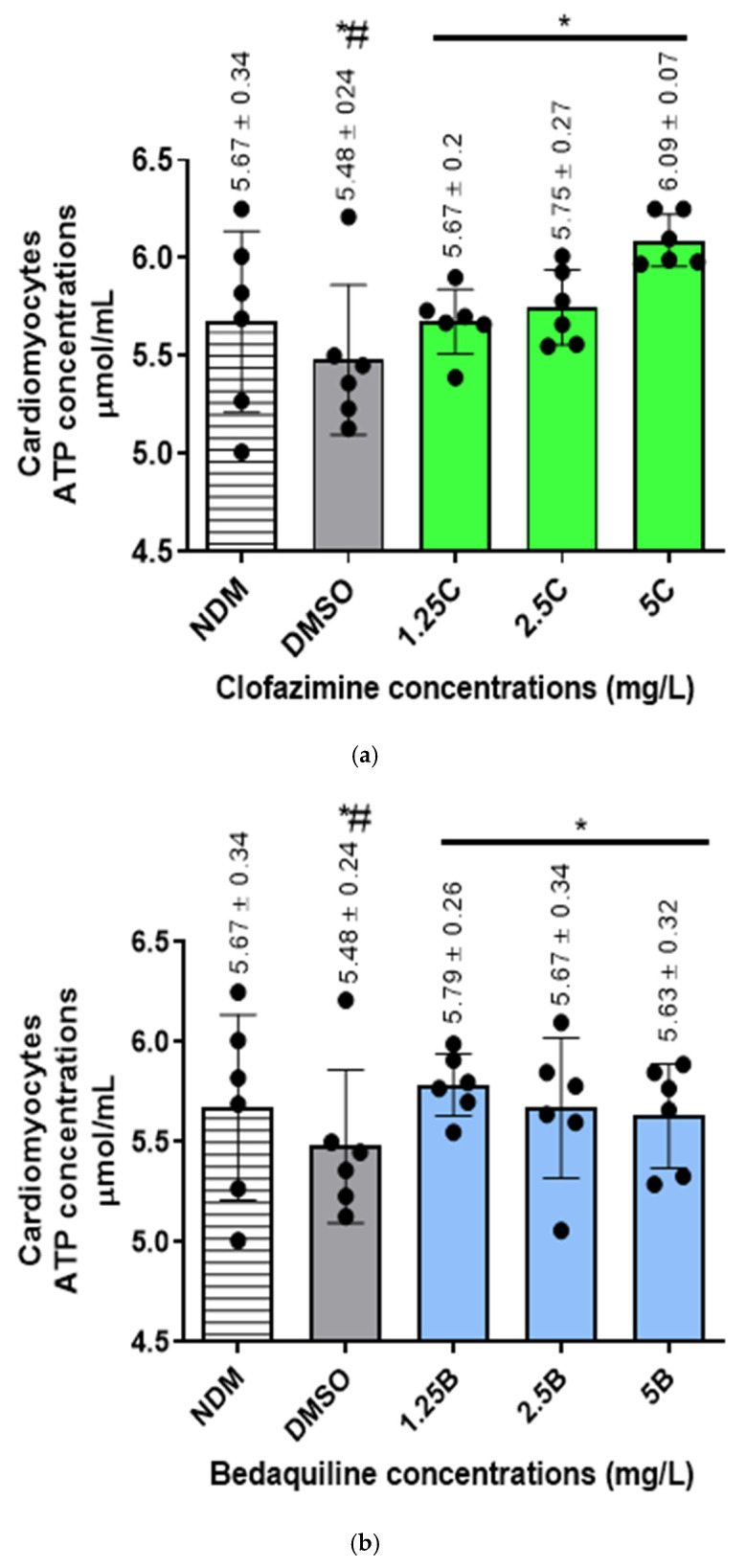
Intracellular ATP concentrations (µmol/mL) in the RCMs treated with various concentrations of (**a**) CFZ and (**b**) BDQ alone and (**c**) in combination. Statistical significance, representing *p* ≤ 0.05, is shown by *. The *p* value representing the statistical difference between NDM and DMSO antibiotic-free controls is denoted by #. The *p* value determined for a statistically significant difference between the NDM and DMSO-treated controls was 0.0102. For the antibiotic treatment assays, these *p* values were: for CFZ = 0.0101, 0.0161, and 0.0022 for 1.25C, 2.5C, and 5C; for BDQ = 0.0065, 0.0241, and 0.0125 for 1.25B, 2.5B, and 5B; for CFZ and BDQ combinations = 0.0022, 0.0299, and 0.292 for 1.25C + 1.25B, 2.5C + 2.5B, and 5C + 5B, combinations, respectively. Abbreviations: ATP, adenosine triphosphate; B, bedaquiline; C, clofazimine; DMSO, dimethylsulphoxide; NDM, no DMSO.

**Figure 3 ijms-25-13022-f003:**
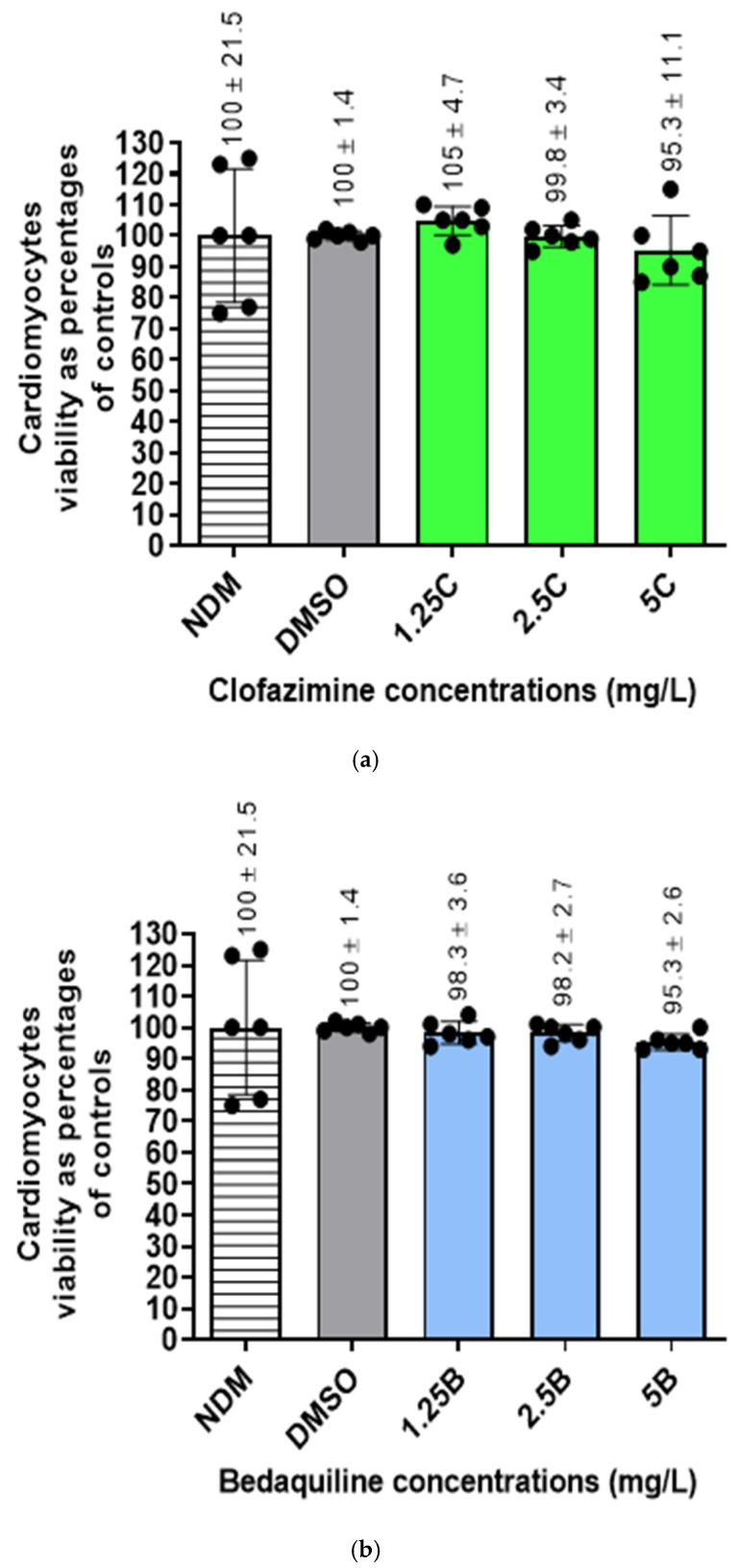
Viability determinations of the RCMs treated with various concentrations of (**a**) CFZ and (**b**) BDQ alone and (**c**) in combination. No statistical difference was shown between the DMSO-treated controls and the antibiotic-treated assays. Abbreviations: B, bedaquiline; C, clofazimine; DMSO, dimethylsulphoxide; NDM, no DMSO.

**Figure 4 ijms-25-13022-f004:**
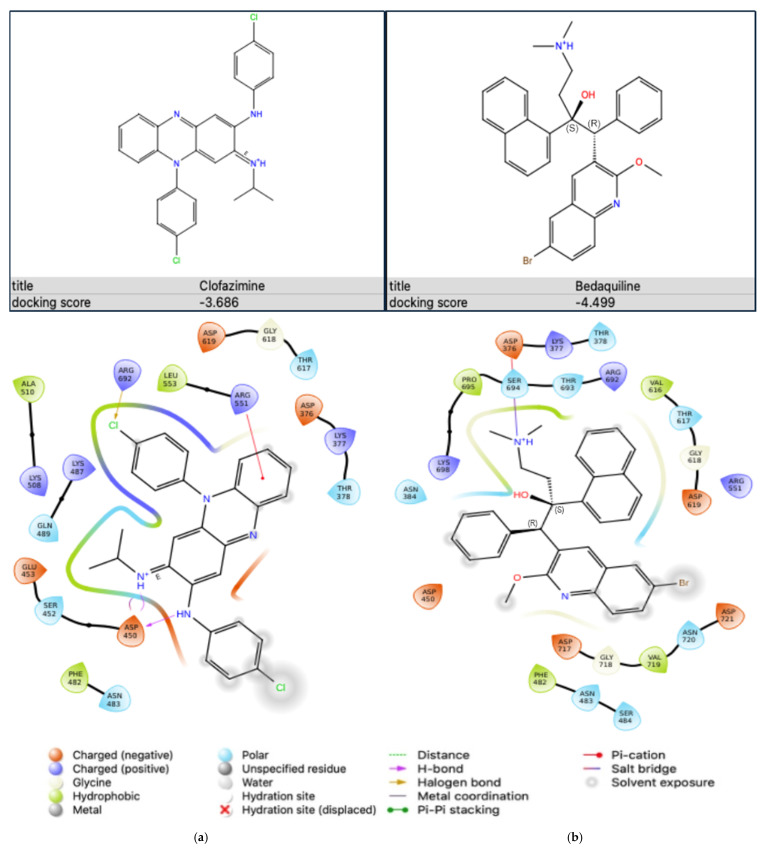
Protein–ligand interaction diagrams obtained from Schrodinger Maestro (Schrödinger Release 2023-2: Maestro, version 13.3) showing (**a**) CFZ bound to PDB ID: 7E21 and (**b**) BDQ bound to PDB ID: 7E21.

**Figure 5 ijms-25-13022-f005:**
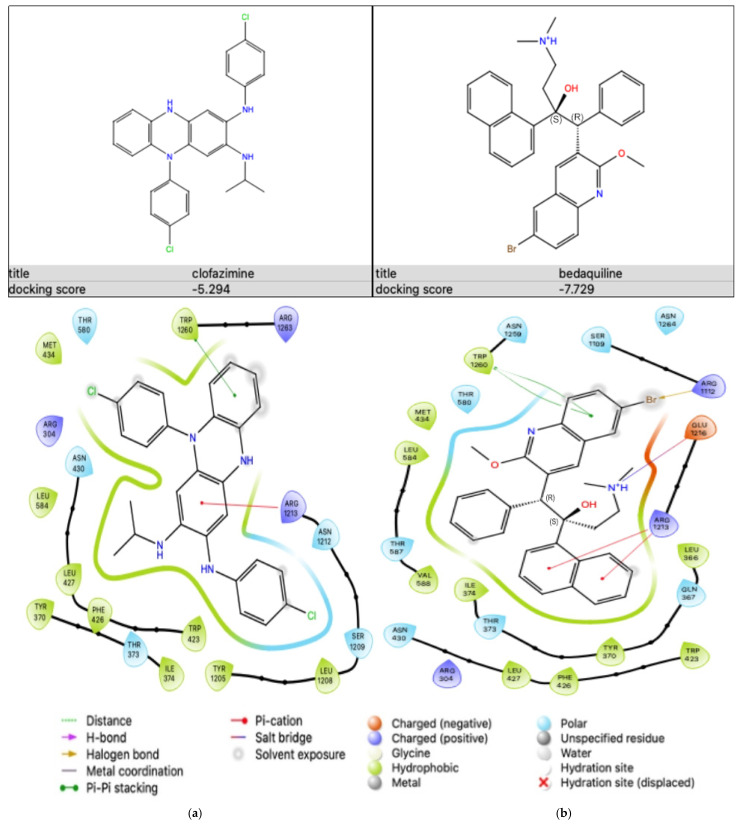
Ligand–protein interaction diagrams obtained from Schrodinger Maestro (Schrödinger Release 2023-2: Maestro, version 13.3) showing (**a**) CFZ bound to PDB ID: 7Y1J and (**b**) BDQ bound to PDB ID: 7Y1J.

## Data Availability

All the datasets generated for this study are included in the article and Appendix A.

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
