# Peer review of "Sodium, Potassium-Adenosine Triphosphatase as a Potential Target of the Anti-Tuberculosis Agents, Clofazimine and Bedaquiline"

_ijms, 2024, doi:10.3390/ijms252313022_

Round 1
Reviewer 1 Report
Comments and Suggestions for Authors
In this paper the authors try to demonstrate the inhibitory effects of CFZ and BDQ individually and in combination, on the activity of the Na+,K+-ATPase pump of the RCMs highlight the existence of additional mechanisms of QT prolongation by these antibiotics. However, although apparently changes in this pump has been found the physiological consequences has not been demonstrated, changes found in ATP concentrations are not exclusively related with this (avoiding essential cellular elements such as mitochondria and other currents such as IKatp, among several others), the presence of other essential experiments such as IKatp currents, Seahorse experiments or glycolytic activity, are lack… In addition, some changes would improve the manuscript: 1) scatter-plots (instead of bar plots to directly visualize the number of experiments performed and their dispersion); 2) the specification of neonatal/young/adult RCMs would be essential to properly understand the experimental setting; 3) avoiding to erase other essential potassium currents (Ito, IKs, IKatp or IK1) among the K+-uptake transporters…
Reviewer 2 Report
Comments and Suggestions for Authors
In this article the authors report some interesting results on the activity of two anti-tuberculosis agents, clofazimine and bedaquiline toward Na+ ,K+ -ATPase pump. The resulting evidences are important for evaluationf of myocardial side-effects such as QT prolongation. The manuscript is well written and just some minor adjustements are necessary before acceptance (see below).
As major drawback, the interaction of the two drugs, BFZ and DBQ, should be also addressed with computational studies, at least with docking, in order to improve the quality of the manuscript and better support the proposed inhibitory activity.
As minor concerns:
1) please improve the quality of figures 1-3, they seems quitely distorted.
2) please use the correct number of significant digits, in some cases there are two decimals and in other three. The same for the corresponding experimental errors.
3) even if concentrations of antimicrobial drugs are usually expressed as mg/L, please also add the corresponding concentration values as molarity, at least into the experimental part.
4) Please add explanation for abbreviations at their first appearence in the text, (for example NDM), while other seems missing (i.e. PI, QT)
5) Please check the standard curve of figure 4S and report the R value.
6) Did the authors evaluate also expression of Na+ ,K+ -ATPase pump by means of WB? This experiment could be also of interest.
Round 2
Reviewer 1 Report
Comments and Suggestions for Authors
Dear authors,
Although some improvements have been performed in the manuscript, there are still some issues that have not been experimentaly adressed (IKatp currents, Seahorse experiments or glycolytic activity...) to support the conclusions
Reviewer 2 Report
Comments and Suggestions for Authors
The authors made requested changes. The manuscript can be accepted.
Author Response
There were no further comments and all the changes have been accepted.
Round 3
Reviewer 1 Report
Comments and Suggestions for Authors
I want to thank the authors for their effort in replying to my concerns. From this reviewer's point of view, this additional work increase the robustness of the work and the potential side effects in the physiology of the cardiomyocytes that might be explored in future projects.
Author Response
The comments that were previously raised have been accepted. There were therefore no further comments.